# Mixed methods process evaluation of an enhanced community-based rehabilitation intervention for elderly patients with hip fracture

Jessica Louise Roberts,[1] Aaron W Pritchard,[2] Michelle Williams,[1] Nikki Totton,[3] Val Morrison,[4] Nafees Ud Din,[1] Nefyn H Williams[5]

¹School of Healthcare Sciences, Bangor University, Bangor, UK
²Research and Development Department, Betsi Cadwaladr University Health Board, Bangor, Gwynedd, UK
³School of Health and Related Research, Sheffield University, Sheffield, UK
⁴School of Psychology, Bangor University, Bangor, UK
⁵Department of Health Services Research, University of Liverpool, Liverpool, UK

**Correspondence to**
Dr Jessica Louise Roberts;
j.l.roberts@bangor.ac.uk

## ABSTRACT

**Objectives** To describe the implementation of an enhanced rehabilitation programme for elderly hip fracture patients with mental capacity, in a randomised feasibility study compared with usual rehabilitation. To compare processes between the two and to collect the views of patients, carers and therapy staff about trial participation.

**Design** Mixed methods process evaluation in a randomised feasibility study.

**Setting** Patient participants were recruited on orthopaedic and rehabilitation wards; the intervention was delivered in the community following hospital discharge.

**Participants** Sixty-one older adults (aged ≥65 years) recovering from surgical treatment (replacement arthroplasty or internal fixation) following hip fracture, who were living independently prior to fracture and had mental capacity and 31 of their carers.

**Interventions** Usual care (control) or usual care plus an enhanced rehabilitation package (intervention). The enhanced rehabilitation consisted of a patient-held information workbook, goal-setting diary and up to six additional therapy sessions.

**Process evaluation components** Recruitment of sites and rehabilitation teams, response of rehabilitation teams, recruitment and reach in patient and carer participants, intervention delivery, delivery to individuals, response of individual patients to the enhanced intervention or usual rehabilitation, response of carer participants, unintended consequences and testing intervention theory and context.

**Results** Usual rehabilitation care was very variable. The enhanced rehabilitation group received a mean of five additional therapy sessions. All of the returned goal-setting diaries had inputs from the therapy team, and half had written comments by the patients and carers. Focus group themes: variation of usual care and its impact on delivering the intervention; the importance of goal setting; the role of the therapist in providing reassurance about safe physical activities; and acceptability of the extra therapy sessions.

**Conclusions** Lessons learnt for a future definitive RCT include how to enhance recruitment and improve training materials, the workbook, delivery of the extra therapy sessions and recording of usual rehabilitation care.

**Trial registration number** ISRCTN22464643; Post-results.

### Strengths and limitations of this study

► Mixed methods process evaluation of a phase II randomised feasibility study, which has examined recruitment, reach, dose delivered, intervention fidelity, unintended consequences, contextual factors and underlying theory.

► It will inform the delivery of a future, definitive, phase III randomised controlled trial.

► It is not possible to comment on the longer term implementation of the enhanced rehabilitation intervention, because this process evaluation was of a feasibility study with only a 3-month follow-up.

► The participants did not include people with severe cognitive impairment, as the lack of mental capacity was an exclusion criterion.

► Despite being a feasibility study with a small sample size, it was possible to gather some evidence supporting the underlying theory with regard to the importance of self-efficacy.

## BACKGROUND

Proximal femoral fracture, known as hip fracture, is a major health problem in the elderly,[1] associated with a reduced ability to conduct activities of daily living independently.[2] Guidelines from the National Institute of Health and Care Excellence recommend the use of multidisciplinary rehabilitation programmes to maximise patient's recovery potential,[3] but there is insufficient evidence of effectiveness and cost-effectiveness, and the importance of individual components of these programmes in achieving desirable outcomes is poorly understood.[4–6]

### Study context

A study funded by the Health Technology Assesment programme[7] completed the first two phases of the MRC framework for complex interventions.[8] The first phase developed an enhanced rehabilitation intervention from the following working theory[9]:

In the context of patients with a great range and variety of pre-fracture physical and mental comorbidities affecting their ability to meet rehabilitation goals, a tailored intervention incorporating increased amount of high quality practice of exercise and activities of daily living leads to better confidence, mood, function, mobility and reduced fear of falling.

In addition to usual care, the intervention included:
► Six home-based therapy sessions delivered by physiotherapists (PTs) or occupational therapists (OTs) with the assistance of technical instructor (TI) providing reliable and consistent care.
► A novel, patient-held, workbook containing information on hip fracture, what to expect from rehabilitation, information about their role in their recovery, importance of physical activity and maintaining functional activities and signposting to other services. The workbook contained a page of questions and Likert scale-type response options to encourage participants to provide feedback on their workbook.
► A diary to facilitate patient-led goal setting, promote engagement and increase self-management.

A logic model described how the intervention components related to the programme theory.[10]

The second phase of the study was a randomised feasibility study, which assessed the acceptability of the new rehabilitation programme and the feasibility of trial methods.[10 11] Participants in the feasibility study were recruited from three acute hospitals across North Wales: East, Central and West. The rehabilitation intervention was delivered in the community. Participants were adults aged 65 years or older who had received surgical treatment for hip fracture, had been living independently prior to the hip fracture, had mental capacity as assessed by their clinical team and received rehabilitation in the North Wales area. Between June 2014 and March 2015, 61 participants were randomised to usual care (control) or usual care plus the enhanced rehabilitation package (intervention). The mean Abbreviated Mental Test Score for these participants was 9.1 (SD 1.3 and range 5–10). A score of 8 or less suggests cognitive impairment.[12]

In addition, participants recruited to the feasibility study were compared with an anonymised cohort of all patients admitted to the same acute hospitals with a proximal femoral fracture over a similar time period. Compared with this cohort, the study participants were younger, less likely to be readmitted to hospital and less likely to die. Outcomes were measured at baseline and at 3-month follow-up and included: disability, activities of daily living, anxiety and depression, health utility, health service resource use, hip pain intensity, self-efficacy, fear of falling, physical function and carer strain.

Guidance from the UK Medical Research Council for developing and evaluating complex interventions recommends conducting a process evaluation, in order to 'explain discrepancies between expected and observed outcomes, and to provide insights to aid implementation'.[8]

This process evaluation aims to describe the implementation of an enhanced rehabilitation programme for elderly hip fracture patients with mental capacity. The specific objectives were to:
► Describe the implementation of the enhanced rehabilitation programme in the intervention group and usual rehabilitation in the control group.
► Describe and compare processes between the two forms of rehabilitation.
► Collect data from trial participants (patients, carers and therapy staff) about their experience of taking part in the trial.
► Collect data about contextual factors and test the theory underlying the intervention.

## METHODS

The study was influenced by Steckler and Linnan's process evaluation framework[13] and other proposed frameworks for designing and reporting process evaluations,[14–16] other process evaluations of trials of complex interventions[17 18] as well as realist evaluation[19] (table 1).

Mixed methods were used to collect process data. Routinely collected electronic health records using Therapy Manager software were used to extract usual rehabilitation activity data for participants in both intervention and control groups. The content of the additional enhanced rehabilitation sessions were recorded by therapy staff onto case report forms. These described how the sessions were used for each patient, including the length of the session, where the session was delivered and the type of activities undertaken (online supplementary appendix 1). Workbooks and goal-setting diaries were collected from patients at follow-up and examined for degree of completion. Questionnaires completed by patients and carers contained health service resource use data. We used descriptive statistics to compare recruitment rates between the sites and to describe the rehabilitation components used from the case report forms, routinely collected electronic records and the completed workbooks and diaries.

We tested the theory underlying the intervention by testing the correlation between the main outcome measure (Barthel Index[20]) and three process measures of self-efficacy: General Self-Efficacy Scale (GSES),[21] Falls Self-Efficacy Scale – International (FES-I)[22 23] and Self-Efficacy for Exercise (SEE).[24] The Barthel Index and GSES were collected at both baseline and follow-up, but FES-I and SEE were only completed at 3-month follow-up.

After the intervention was completed, we carried out focus group interviews of patient and carer participants (topic guide in online supplementary appendix 2). Separate focus groups were conducted for those in the control and intervention groups. Healthcare staff who delivered the intervention were also invited to separate focus groups at their nearest acute hospital site. Where staff were unable to attend, one-to-one telephone interviews were offered. Focus groups were recorded, transcribed and analysed thematically by two researchers. Patient and

**Table 1** Process evaluation questions and methods for evaluating

| Component | Process evaluation questions | Research methods | Stage of study to collect data |
|---|---|---|---|
| Recruitment of sites and rehabilitation teams | How are sites and teams recruited? | Documentation of recruitment process by research team. | Preintervention |
| | Which sites and teams agree to participate? | Quantitative comparison of recruited and non-recruited sites. | |
| Response of rehabilitation teams | How is the enhanced intervention adopted by the rehabilitation teams? | Quantitative examination of case report forms and qualitative interviews of rehabilitation team members. | During and following the intervention |
| Recruitment and reach in patient and carer participants | How many are recruited into the feasibility study? Are they representative? | Quantitative comparison between feasibility study and anonymised cohort. | During the intervention |
| | Who is recruited into the feasibility study? What are the reasons for non-recruitment? | Examination of recruitment log. | During the intervention |
| Intervention delivery | What rehabilitation intervention is delivered? Is it what was intended by the researchers? | Quantitative examination of case report forms and of electronic data entered onto Therapy Manager software. | During the intervention |
| Delivery to individuals | What intervention is delivered to each participant? | Quantitative examination of case report forms and of electronic data entered onto Therapy Manager software. | During the intervention |
| | Is the delivered intervention the one intended by the researchers? | Measurement of intervention fidelity: completion of workbook tasks, completion of diaries and number and content of therapy sessions. | During the intervention |
| Response of individual patients to the enhanced intervention or usual rehabilitation | How do the patient participants respond? | Qualitative analysis of focus group data about patient participants' experience and response to the intervention and to usual care. | Following the intervention |
| Response of carer participants | Effects on carers. | Qualitative analysis of focus group data about carers' experiences. | Following the intervention |
| Unintended consequences | Are there unintended changes in processes and outcomes related to the intervention and unrelated to care? | Quantitative examination of adverse effects reports, health service activity data from patient completed questionnaires and routinely collected electronic sources. Qualitative analysis of focus group data from patients and their carers. | During and following the intervention |
| Theory | What theory has been used to develop the intervention? | Quantitative data analysis of process outcome measures to assess predicted relationships. | Following the intervention |
| Context | What is the wider context in which the feasibility study is conducted? | Realist review of the rehabilitation literature, survey of current rehabilitation practice, focus groups of patients, carers and rehabilitation professionals. Quantitative comparison with anonymised cohort. | Preintervention (phase I study) and during the intervention |

carers were asked about their experience of rehabilitation in general and of taking part in the Fracture in the Elderly Multidisciplinary Rehabilitation (FEMuR) study.

## PATIENT AND PUBLIC INVOLVEMENT
Patient and public involvement representatives were involved in the development of the original application for funding of the feasibility study and provided input on the choice of outcome measures, content of the intervention documents and patient facing materials. Topic guides for the focus groups were developed iteratively following feedback from early focus group participants. Participants who requested information on study findings were sent an overview of the results and invited to input into the development of the future definitive trial. The burden of the intervention to patients and their carers was discussed in the focus groups and formed an important part of assessing acceptability of the intervention.

## RESULTS
### Recruitment of sites and rehabilitation teams
Rehabilitation team leads at three acute hospitals identified PTs based on the acute orthopaedic wards and OTs and dual-trained TIs who were both acute and community based. The structure of the teams trained to deliver the intervention at each site differed depending

**Table 2** Eligibility, recruitment and retention rates according to acute hospital site

| Number of Patients | West | Central | East | Total |
|---|---|---|---|---|
| Screened | 147 | 235 | 211 | 593 |
| Eligible (rate %) | 75 (51) | 103 (44) | 88 (42) | 266 (45) |
| Recruited (rate %) | 11 (15) | 35 (34) | 16 (18) | 62 (23) |
| Retained (rate %) | 4 (36) | 29 (83) | 16 (100) | 49 (79) |

on staff availability, with at least one Band 6 PT at each site who led the teams. As the PTs in West and Central were based only within the acute hospital, the initial assessment session and introduction of the hip fracture workbook took place in the acute setting. In East, the PT was able to conduct this session with the patient in the community following discharge. Rehabilitation teams were advised to support the patients in setting individual goals that could be worked on in the intervention sessions, with a particular focus on activities of daily living. Specific content of the session was decided at the discretion of the therapist and was dependent on individual patient need. Training sessions also included information on how to screen potential participants and how to complete intervention paperwork to capture detail on how additional sessions were used.

### Response of rehabilitation teams
The initial recruitment period was planned to last 6 months. Due to staffing difficulties and the rurality of the West area, there were challenges delivering the intervention, and recruitment was slower than expected. Recruitment was extended for 3 months in Central and East but was closed in the West.

### Recruitment and reach in patient and carer participants
Rates of recruitment, eligibility and retention are given in table 2. The main reasons for ineligibility were: lack of mental capacity 161 (49%), not living independently 61 (19%), younger than 65 years (13%), living out of area 30 (9%) and treated without surgery 23 (7%). Patients were recruited after 193 (73%) eligible patients were approached with 176 (91%) of these agreeing to talk to the researcher. Those who were not approached had either been: discharged home before recruitment, died, lived in areas where it was not possible to deliver the intervention, were deemed by clinical staff to be too ill to take part or there were safety concerns that would have prevented the intervention being delivered due to lone worker policies. The main reasons for lack of recruitment in those approached were: burdensome 60 (31%) or disliked the study or questionnaire 13 (7%). Information concerning the number of visits it took to recruit participants were collected for 36 patients. The majority of patients had two visits, because recruitment occurred early in patients' recovery from surgery, and many requested a return visit to discuss the study after they had been discharged.

The retention rate was highest in the East and lowest in the West. The West encountered particular difficulties accessing staff for the trial, which might explain their poor retention rate. Nine patients withdrew from the study and four could not be contacted, so they were considered lost to follow-up.

In addition, 41 carers were identified and 31 (75%) agreed to participate. Six carers withdrew from the study, and seven were lost to follow-up, leaving 18 (58%) who completed the follow-up questionnaire.

### Intervention delivery
Data describing usual therapy care were only available from 35 participants recruited in the Central hospital and associated community therapy teams, who were using Therapy Manager software. Five of these participants withdrew from the study, and no further data regarding usual care were collected. Following discharge from the acute hospital, patients were discharged to their place of residence or for further rehabilitation in a community hospital prior to going home (online supplementary appendix 3). Ten patients had no details recorded relating to usual care following acute hospital discharge. Of the 20 patients who did have entries, four did not receive any face-to-face appointments with a healthcare professional, as their entries related to telephone calls to patients who were either uncontactable or declined further treatment. For the 16 patients who received an appointment, there was a median of three appointments (n=4). The maximum number of appointments for one patient was 21. There was a total of 81 appointments for these 16 patients with 73 of these appointments (90%) conducted as home visits. Home visits were completed by different members of the therapy team (online supplementary appendix 4). If an assessment was required, then a qualified PT or OT completed the visit, while subsequent visits following an agreed care plan were conducted by a TI. Most (90%) outpatient appointments were conducted by a PT (10% not recorded).

Activities in these usual rehabilitation sessions were categorised by the researcher as direct or indirect. Direct activities involved the practice of activities of daily living (25%), physical exercise (23%), phone calls with patients, discussion of progress and assessment of mood. Indirect activities were predominantly referrals to other services (33%) or contact with other members of the multidisciplinary team (30%).

Therapy Manager also recorded qualitative data. A number of patients were reluctant to engage in physical activity until they had been seen by a PT, even though in many cases they were told there would be a wait of at least 4 weeks.

Twenty-nine people were randomised to the enhanced rehabilitation intervention, and details were available for 20 (reasons for missing data in online supplementary appendix 5). The majority (n=13) received all six sessions. The mean number of sessions delivered was 4.7 (SD 1.6, range 1–6). One patient randomised to the intervention

was discharged from the community hospital to a respite care home, so her intervention therapy sessions were delivered there.

TIs conducted the majority (55%) of intervention sessions, with 44% conducted by PTs and the remaining 1% by more than one team member. The content of the intervention sessions depended on individual patient need, at the discretion of the treating therapist. Therapists consistently completed the intervention paperwork detailing the types of activities and the time taken. Each session lasted approximately 1 hour, with an additional hour for travelling. In the intervention sessions, there was a lower rate of practising exercises (15%) and activities of daily living (14%) than usual care. Instead, there was more answering questions raised by the intervention workbook, working with goal setting diaries, giving feedback on progress and discussing emotional needs. For indirect activities, only 7% was used for discussion with the wider team and 4% for referring to other services. The remaining indirect activities included travel to appointments, writing notes, arranging further appointments and discussions with carers.

### Delivery to individuals
Ten participants returned their goal-setting diaries and workbooks to the study team. All of the diaries had inputs from the therapists detailing the goals that were set in the initial assessment session. Five had also been updated by patients and their carers. These participants used the diaries extensively, updating their progress on the initial goals agreed and entered by the therapist and including new goals, which they had entered into the diary themselves. Three of these participants also completed quizzes and hip fracture story sections of the workbook. One of the workbooks was completed by a carer who described the challenges to the patient's recovery and what they were doing to overcome them.

### Response of individual patients to the enhanced intervention or usual rehabilitation
Four focus groups were conducted with patients and carers and two with healthcare professionals involved in the intervention (table 3). Due to the geographical spread of participants in the West, it was not possible to conduct a focus group in this area, although one participant from this area was able to attend a focus group in Central. Healthcare professionals delivering the intervention in the West were unable to attend focus groups, but one acute PT and three TIs participated in individual telephone interviews. Four themes emerged:

#### Theme 1: variation of usual care and its impact on delivering the intervention
The frequency and format of usual community rehabilitation varied because of tailoring to individual need, the availability of resources and the provision of support services such as reablement and falls prevention classes. One carer described this variation as a '*postcode lottery*'

**Table 3** Focus group participants' characteristics

| Participant type | Location | Attendees |
|---|---|---|
| Patient and carers in control group | East | Two female patients, one male patient and two male carers (n=5). |
| Patient and carers in control group | Central | Two female patients, one male patient and one female carer (n=4). |
| Patient and carers in intervention group | East | Three female patients (n=3). |
| Patient and carers in intervention group | Central | Two male patients, two female patients, one male carer and two female carers (n=7). |
| Healthcare professionals | East | Clinical specialist physiotherapist, two orthopaedic physiotherapists and physiotherapy technical instructor (n=4). |
| Healthcare professionals | Central | Orthopaedic acute physiotherapist, rotational physiotherapist and physiotherapy technical instructor (n=3). |

(male carer, control group). In the control group, the initial contact with therapists often needed to be initiated by the patient, relying on their self-motivation, which was not necessary for the intervention group.

According to therapists, there was large geographical variation in usual care ranging from multiple same day appointments to no rehabilitation whatsoever. This variation affected how the enhanced rehabilitation intervention was delivered. One therapist commented that when she delivered the intervention to patients with minimal usual care she would '*spread out the sessions, and then just pushed [the patient] harder, in the two weeks*' (physiotherapist). In contrast, where a comprehensive rehabilitation programme was available, she would deliver intervention sessions weekly in the confidence that at the end of the intervention period, this provision would be continued through community-run falls prevention or exercise schemes.

Therapists were concerned about supporting patients to set individualised goals when these might conflict with goals supported by other rehabilitation providers.

> It was much harder when they had [another ongoing service], the re-ablement ones were much harder to actually, because somebody else was already setting what they were going to achieve. (Physiotherapist)

#### Theme 2: the importance of goal setting
Goal setting was identified by therapists as playing an important role in engaging patients with their own

recovery and in providing motivation to regain function and independence.

> I think [patients] probably more motivated because they can see the steps, to getting to that point. And why you are doing it. (Physiotherapist)

The patient-held goal setting diary were appreciated by participants, as it gave them a direct focus and accountability for their goals.

> You feel as if you have got a goal to get to, because you have put it in that book and you have got a goal. (Female patient, intervention group)

Therapists felt that the workbook and diary enabled patients to be more involved in their rehabilitation.

### Theme 3: the role of the therapist in providing reassurance about safe physical activities

The majority of patient participants reported anger and frustration when their physical ability to progress did not match their expectations, and they remained dependent on others.

> Being incapacitated infuriated me so much. (Female patient, intervention group)

It was at this point that the PT played a pivotal role in managing expectations and reassuring patients that they were progressing normally. In the absence of this support, there was a risk of patients losing motivation. This reassurance was important for giving patients the confidence to perform physical activities, as there was an underlying concern that they may otherwise do exercises that may be harmful.

> I had the security to know they were the right exercises, somebody there who gave them to me and you know they are qualified and they are telling you the right thing to do. (Female patient, control group)

For participants in the control group, this lack of reassurance was a particular problem. Patients received a list of activities to avoid (hip precautions), but some were given no information about what exercises and activities were safe to perform, and wanted access to:

> [S]omebody I could have just picked up the phone and said, how about this, should this be happening. (Female patient, control group)

Both groups identified this initial contact with therapists as vital for building their confidence and supporting their self-motivation for recovery.

> Once you have the information and the guidance on what to do, what not to do, I think we are intelligent enough to go away and do it, but it is just that initial guidance… we might be capable but you still need guidance. (Male patient, control group)

Patients emphasised the importance of the intervention sessions in allowing them time to discuss their individual problems, particularly in the early stages of rehabilitation. This was facilitated by their relationship with the therapist or therapy team, where they felt comfortable enough to ask questions without fear of being dismissed or considered a nuisance. This was in contrast with how they felt in the acute hospital, or in usual care, where they were less informed about the processes and unfamiliar with the staff. A good relationship with their therapist underpinned successful rehabilitation and enabled them to engage in, and take responsibility for, their role within the recovery process.

> I felt as though it was a sort of team effort, and she [the therapist] was sort of team leader, and knew what to do, and then it is sort of from part of the team if you like. (Male patient, intervention group)

### Theme 4: acceptability of the extra therapy sessions

Patient feedback on the intervention workbook varied. Some patients appreciated the explanation of the mechanics of their fracture and their better understanding of the surgery used to fix it.

> I thought it was good because it did explain things, it did explain to you what happens with a fracture. (Female patient, intervention group)

Other patients reflected on the comfort that this additional information gave them.

> I didn't know what to expect but I found I read [the workbook] profusely every day, and I did, I found it very, very helpful. It made me feel that I wasn't on my own. (Female patient, intervention group)

Other participants found the workbooks less useful.

> I sort of read it once and thought well you know this isn't very useful. (Male patient, intervention group)

Without exception, the most useful aspect of the intervention was the extra time that participants received with the therapy teams. The goal-setting diary and information workbook were seen as useful supporting documents to these extra sessions. While therapists acknowledged the complex nature of delivering intervention sessions in an environment of varied usual care, it was generally accepted that the extra sessions were a great benefit to patients. The analysis of the focus groups also led to the development of the GUIDE tool (figure 1), which summarises the role of the therapist in the rehabilitation process, incorporating important factors identified by patients and their carers.

### Unintended consequences

There were nine adverse events, six were serious, two resulted in readmission to the acute hospital and there was one death; none were related to participation in the study.

### Testing the intervention theory

Correlations between the Barthel Index and the three process measures of self-efficacy were statistically

ROLE OF THE PHYSIOTHERAPIST

**G** UIDANCE on self-directed/community supported rehabilitation to reduce common anxieties e.g. of falling.

**U** NDERSTANDING concerns and information needs of patients through a model of one-to-one support.

**I** NDIVIDUALISED support responsive to individual goals, co-morbidities, level of mobility and need.

**D** IRECTION on achieving self-set goals and recovery expectations with individualised strategies.

**E** XPERTISE working within an evidence based framework to enable post-surgical recovery at home and in the community

**Figure 1** GUIDE mnemonic for therapists involved in rehabilitation following hip fracture.

significant and suggested that higher levels of activities of daily living were associated with higher scores of self-efficacy (online supplementary appendix 6). Similarly, higher scores in the FES-I represent a greater fear of falling, which was associated with lower levels of activities of daily living. The strongest correlation was with the FES-I.

## DISCUSSION
### Summary of main findings
This study took place in three sites across North Wales. Recruitment to the study was more difficult in the West because of its rurality and also staff shortages. The recruitment rate was highest in Central; the retention rate was highest in the East. Usual rehabilitation care was very variable with a median of three appointments; the enhanced rehabilitation group received a mean of five additional therapy sessions. Variation in usual care affected how the enhanced rehabilitation intervention was delivered. TIs carried out most of the sessions, which consisted of practising exercises and activities of daily living, goal setting, answering questions raised by the workbook and giving feedback on progress. Goal setting had an important role in engaging patients in their own recovery, which was assisted by the workbook and diary. All of the returned goal-setting diaries had inputs from the therapy team, and half had written comments by the patients and carers. Some participants did not find the workbook and diary useful, but all valued the extra therapy sessions. The PT was very important for managing patients' expectations and for reassuring them about what physical activity was safe to perform. The lack of reassurance was particularly problematic for some in the control group. There were statistically significant correlations between three process measures of self-efficacy and the Barthel index, which supported the theory underlying the intervention.

### Strengths and weaknesses
This was a mixed methods process evaluation performed concurrently with a randomised feasibility study that has examined recruitment, reach, dose delivered, intervention fidelity, unintended consequences, contextual

factors and underlying theory. Rates of recruitment and retention were low, and this process evaluation informs trial methods, as well as how to deliver the intervention for a future, definitive, phase III RCT. Although many commented that goal setting was enhanced by the workbook and self-monitoring diaries, half of the collected workbooks were not completed, and some found them unhelpful. Feedback from participants and intervention delivery staff will result in further refinement of the workbook and diary and also the development of training materials before the definitive trial. Because this process evaluation was only part of a feasibility study, it is not possible to comment on longer term implementation of the enhanced rehabilitation intervention. The process data were analysed concurrently with the outcome data from the feasibility study, so the analysis of quantitative data was performed blind to treatment allocation; however, the qualitative findings and the feasibility study outcomes were discussed in relation to one another. It was not possible to collect data on usual rehabilitation from all participants in the intervention group, because the Therapy Manager software was only used by the rehabilitation teams in the Central area.

The participants did not include people with severe cognitive impairment, as the lack of mental capacity to give informed consent was an exclusion criterion. However, people with milder cognitive impairment, but who still had mental capacity, were not excluded.

The enhanced rehabilitation intervention was delivered by PTs and TIs, with very little input from OTs. This was due to the availability of PTs and the shortage of OTs in this health board during the study period. We believe

| Table 4 | | Ten lessons learnt for a future definitive randomised controlled trial |
|---|---|---|
| Sites | 1 | Consider staff availability and rurality when recruiting sites. |
| Therapy staff | 2 | Consider employing therapy staff directly or second from research delivery teams. |
| Participant recruitment | 3 | Recruitment flexibility, including after discharge home. |
| | 4 | Keep visiting potential participants. |
| | 5 | Delay recruitment until later after surgery. |
| Trial-specific training | 6 | Use the mnemonic GUIDE as a training aid. |
| | 7 | Stress the importance of the first home visit to reassure participants about safe activities. |
| | 8 | Value the emotional support provided by the technical instructors. |
| Usual care recording | 9 | Patient-held treatment log completed by visiting therapists. |
| Workbook | 10 | Further refine the workbook in the light of feedback. |

that there is sufficient overlap in rehabilitation practice for these findings to be relevant to OTs as well. Also, the TIs who delivered most of the extra rehabilitation sessions worked with both PTs and OTs. The extra rehabilitation sessions concentrated on improving self-efficacy and personal goal setting more than the practice of exercise and Activities of Daily Living (ADLs). We did manage to capture the practice of exercise and ADLs in participants' own time in a small number of participants who returned their diaries; however, we do not know how often participants in the control group practised their exercises and ADLs outside of therapy sessions.

Despite being a feasibility study with a small sample size, it was possible to gather some evidence supporting the underlying theory with regard to the importance of self-efficacy.

### Comparison with previous literature
Qualitative interviews of participants in the Exercise-Plus RCT in the USA, of a motivational intervention designed to increase adherence to rehabilitation exercise, also found that identifying goals and improving self-efficacy were important, and an exercise booklet provided useful visual cues.[25] A good relationship with the therapist providing individualised care and verbal encouragement resulted in participants reciprocating their therapists' kindness by completing the exercise programme. They also described constraints to exercise such as unpleasant sensations of pain and fatigue, lack of time and space, and discontinuing their exercise once baseline function was restored. A qualitative study of a rehabilitation programme in Taiwan found that when therapists emphasised social support and resilience, patients developed more self-confidence and independence.[26] A process evaluation of a rehabilitation intervention in Sweden found that hip fracture had long lasting 'social and existential' effects on patients necessitating both physical and emotional support during recovery.[27] The recent 'Hip Sprint' audit reviewed physiotherapy rehabilitation for hip fracture patients throughout the UK.[28] It found that usual rehabilitation care was variable with wide variation in the delay before home rehabilitation started, the amount and frequency of visits and the type of staff involved. PTs, and in particular PT assistants, provided most of the care.

### Implications for a future definitive RCT
Several lessons have been learnt for delivering the enhanced rehabilitation intervention to elderly hip fracture patients with mental capacity in a definitive phase III RCT (table 4). Recruitment was harder in rural areas, especially in areas with staff shortages, which will be an important consideration when choosing sites for the definitive trial. Research staff need to remain flexible, be prepared to recruit after discharge home, keep visiting potential participants and possibly delay recruitment until later after surgery. Employing therapists directly by the research team or secondment from the research delivery workforce would avoid them from being pulled back to clinical work during staffing shortages. The workbook and goal-setting diary need to be refined further in the light of feedback from patients, carers and clinicians. A mnemonic (GUIDE) for therapists has been developed following the qualitative research (figure 1), which will be useful as a training tool for the therapy teams prior to a definitive phase III RCT. The collection of usual rehabilitation care data could be enhanced by using a patient-held treatment log completed by visiting therapists.

**Acknowledgements** The FEMuR team would like to thank all participants who took part in the study. The authors would also like to thank Health and Care Research Wales for supporting participant recruitment and the healthcare staff at Betsi Cadwaladr University Health Board who identified potential participants and delivered the intervention.

**Contributors** NHW was the chief investigator and grant holder, was responsible for study design, conduct and analysis and had overall responsibility for the study and acts as guarantor. JLR was the study manager overseeing day-to-day conduct, participant recruitment and methodological input and conducted qualitative analysis for the focus groups. NUD was involved in participant recruitment, acquisition of quantitative and qualitative data and analysis. NT conducted the statistical analysis for the feasibility study. Her institution affiliation changed from Bangor University to the University of Sheffield during the study. VM was a coinvestigator responsible for study design, provided health psychology expertise and methodological oversight throughout the study. MW was involved in the conduct of the study including maintenance of study documentation and acquisition of data and provided administrative support. AWP contributed to qualitative analysis of the focus groups. All authors were involved in drafting, revising and approving this manuscript.

**Funding** This work was supported by the National Institute for Health Research's Health Technology Assessment Programme, grant number 11/33/03.

**Disclaimer** The views and opinions expressed therein are those of the authors and do not necessarily reflect those of the HTA, NIHR, NHS or the Department of Health.

**Competing interests** NHW, JLR, NUD, MW, NT and VM report a grant from NIHR HTA programme, for the conduct of the study. NHW reports additional grants from Public Health Wales, NIHR HTA and BCUHB, outside the submitted work.

**Patient consent** Not required.

**Ethics approval** The study received ethical approval from the UK NHS North Wales West Research Ethics Committee—West. Ref 13/WA/0402 and NHS Research and Development approval from the Betsi Cadwaladr University Health Board Internal Review Panel.

**Provenance and peer review** Not commissioned; externally peer reviewed.

**Data sharing statement** The datasets used during the current study are available from the corresponding author on reasonable request.

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
