## [Reviewer comments · BMJ Open]

ARTICLE DETAILS

TITLE (PROVISIONAL)	Mixed methods process evaluation of an enhanced community-based rehabilitation intervention for elderly hip fracture patients.
AUTHORS	Roberts, Jessica; Pritchard, Aaron; Williams, Michelle; Totton, Nikki; Morrison, Val; Din, Nafees; Williams, Nefyn

VERSION 1 – REVIEW

REVIEWER	Lene Lindberg Karolinska Institutet, Sweden
REVIEW RETURNED	13-Feb-2018

GENERAL COMMENTS	This is a well-designed and prepared study that it is a pleasure to read about. I have however some minor questions to the authors. The first one relates to page 6, line 48, where the authors mention a comparison to anonymised cohort, is this cohort also patients with a hip fracture from North Wales? It may be helpful for the reader with a clarification of the cohort. My second question is about training of the teams to deliver the intervention mentioned on page 9, lines 18-19. What kind of training did the teams receive? I may have missed something in the study protocol but I have tried to find some details about training.
---

REVIEWER	Joris Slaets University of Groningen, University Medical Center Groningen, The Netherlands
REVIEW RETURNED	18-Feb-2018

GENERAL COMMENTS	The research question is relevant, but I really wonder whether a RCT is useful in such a complex situation. There is a large variation in almost every variable and almost nothing is under experimental control. My worries are not only about this trial but about the prospect of conducting a similar phase III trial. The aim is to improve health care outcome and because the systems are very different, and the patients are very different I don't see the point of aiming at one standardised intervention. I would suggest the authors to consider using the information from this trial to implement learning health care systems which will be different in different regions and for different patient groups. Probably the first priority will be to increase the recruitment rate for rehabilitation programs and to define a primary outcome which is closely related to the experience of the individual patient. There is hardly any statistic in the paper and I agree with that. The most important information is qualitative in nature. However, in appendix 6 there is a correlation analysis and I don't see how this is a fair test of a theory? The low recruitment and attrition rates need more attention in the
---

	description of the weaknesses of the study.
REVIEWER	Victoria Goodwin University of Exeter, UK
REVIEW RETURNED	06-Apr-2018

GENERAL COMMENTS	Thank you for the opportunity to review this well written and interesting manuscript. Page 9 recruitment of rehab teams. How many of each of the different staff types were trained (PT/OT/TI)? Page 10 line 49: is the average sessions the mean or median? Depending which can you also give SD or IQR as appropriate Page 11. Interesting that 99% of intervention delivered by TI or physio indicating very little (if any) input from OTs. I think this is important moving to the main trial as if the OTs are going to be involved then don't bother training them (hence my point from page 9 above). Also interesting that the content of the intervention was very little exercise/ADL practice. When understanding the mechanism of effect from your logic model if people aren't doing things then the mechanism will differ (unless they are doing them unsupervised between visits but this isn't clear) Page 11: 49% of potential participants lacked capacity and were ineligible. Bearing in mind that a very large proportion of people with #NOF will have cognitive issues, this has implications for generalisability of findings down the line. It would be useful to know the cognitive status of participants recruited into the feasibility study so as to truly understand who the study is feasible with/for. Page 18 first para. As you only measured FES-I cross-sectionally rather than longitudinally can you say that as fear of falling increased as ADLs decreased? Surely it is just that those with higher FoF have lower ADL activity??? ie an inverse relationship between FoF and ADLs – no surprise there though!! General points: I found the results section confusing to read as it kept jumping between recruitment and delivery issues so didn't flow as well as it could. Just needs reordering. If the Therapy manager software didn't really give you a clear picture of usual care across all sites is this going to be feasible moving forwards into the full trial process evaluation to use it. Not sure it is used much in England and I've never come across it before. Being able to clearly describe usual care is important and work we are currently doing suggests that if we can't clearly describe usual care in a trial or it is so variable, it can be difficult to distinguish it from the intervention and may result in a null effect. You may also want to reference the findings of the recent CSP HipSPRINT national audit about usual care following hip fracture I like Box 1 on lessons learned but I think it could be expanded upon to include (a) the types of staff who would be trained to deliver the intervention and (b) alternative methods of collecting usual care data Please add CONSORT form for pilot trials http://www.consort-statement.org/extensions/overview/pilotandfeasibility
---

VERSION 1 – AUTHOR RESPONSE

Reviewers' Comments to Author:

Reviewer: 1

This is a well-designed and prepared study that it is a pleasure to read about.

REPLY: Thank you.

I have however some minor questions to the authors. The first one relates to page 6, line 48, where the authors mention a comparison to anonymised cohort, is this cohort also patients with a hip fracture from North Wales? It may be helpful for the reader with a clarification of the cohort.

REPLY: This has been amended to "The second phase of the study was a randomised feasibility study, which assessed the acceptability of the new rehabilitation programme and the feasibility of trial methods."

Added to P7 line 48. In addition participants recruited to the feasibility study were compared with an anonymised cohort of all patients admitted to the same acute hospitals with a proximal femoral fracture over a similar time period.

My second question is about training of the teams to deliver the intervention mentioned on page 9, lines 18-19. What kind of training did the teams receive? I may have missed something in the study protocol but I have tried to find some details about training.

REPLY: Rehabilitation teams were advised to support the patients in setting individual goals which could be worked on in the intervention sessions, with a particular focus on activities of daily living and practical skills which the patient wanted to work on. The rehabilitation teams responsible for delivering the additional sessions were also delivering the usual care to participants, which was based on individual patient need. Therefore, the rehabilitation teams were instructed to use the additional sessions for those in the intervention group at their own discretion, allowing them to use this time in a way that would be most useful for achieving patient goals. Training sessions also included information on how to screen potential participants and how to complete intervention paperwork to capture detail on how additional sessions were used. This has been clarified in the text. .

Reviewer: 2

The research question is relevant, but I really wonder whether a RCT is useful in such a complex situation. There is a large variation in almost every variable and almost nothing is under experimental control. My worries are not only about this trial but about the prospect of conducting a similar phase III trial.

REPLY: This is a complex intervention and we have used the MRC framework for evaluating complex interventions. The results of the feasibility study suggested that a phase III RCT was feasible, and it has been funded by the NIHR HTA programme.

The aim is to improve health care outcome and because the systems are very different, and the patients are very different I don't see the point of aiming at one standardised intervention.

REPLY: We agree, we have not aimed at one standardised intervention. The enhanced rehabilitation intervention was designed to complement usual rehabilitation care and not to replace it.

I would suggest the authors to consider using the information from this trial to implement learning health care systems which will be different in different regions and for different patient groups.

REPLY: Thank you. In the first instance the results from the feasibility study and this process evaluation have informed a multi-centre phase III RCT, which will be run from different regions in England and Wales.

Probably the first priority will be to increase the recruitment rate for rehabilitation programs and to define a primary outcome which is closely related to the experience of the individual patient.

REPLY: We agree that increasing the recruitment rate is important and we will use the lessons learnt from this process evaluation to maximise recruitment in the planned phase III RCT. The planned primary outcome measure will be the Nottingham Extended Activities of Daily Living scale which is pertinent to the lived experience of patients following hip fracture.

There is hardly any statistic in the paper and I agree with that. The most important information is qualitative in nature. However, in appendix 6 there is a correlation analysis and I don't see how this is a fair test of a theory?

REPLY: The correlation provides some limited support for the underlying theory.

The low recruitment and attrition rates need more attention in the description of the weaknesses of the study.

REPLY: We agree, this has been done.

Reviewer: 3

Thank you for the opportunity to review this well written and interesting manuscript.

REPLY: Thank you.

Page 9 recruitment of rehab teams. How many of each of the different staff types were trained (PT/OT/TI)?

REPLY: The structure of the rehabilitation teams differed at each site, with suitable staff being identified by rehabilitation team leads. In West and East, as many staff as possible attended the training but not all were involved in delivering the intervention. This depended on staff availability at the time of patient recruitment, but was led by a Band 6 Physiotherapist in each site. In Central, it was planned for only three staff members to be involved in delivering the intervention sessions and as such only these three were trained (two physiotherapists and one dual-trained technical instructor) but all other staff on the team were aware of the study.

Page 10 line 49: is the average sessions the mean or median? Depending which can you also give SD or IQR as appropriate Page 11.

REPLY: This is a mean which has been clarified. The SD and range has been added.

Interesting that 99% of intervention delivered by TI or physio indicating very little (if any) input from OTs. I think this is important moving to the main trial as if the OTs are going to be involved then don't bother training them (hence my point from page 9 above).

REPLY: This was due to the availability of physiotherapists and the shortage of occupational therapists in this health board during the study period. We believe that there is sufficient overlap in rehabilitation practice for these findings to be relevant to OTs as well. Also, the technical instructors who delivered most of the extra rehabilitation sessions worked with both physiotherapists and occupational therapists. This has been added to the discussion.

Also interesting that the content of the intervention was very little exercise/ADL practice. When understanding the mechanism of effect from your logic model if people aren't doing things then the mechanism will differ (unless they are doing them unsupervised between visits but this isn't clear)

REPLY: Yes, the extra rehabilitation sessions concentrated on improving self-efficacy and personal goal-setting. We managed to capture the practice of exercise and ADLs in a small number of participants who returned their diaries. However, we do not know how often participants in the control group practised their exercises and ADLs outside of therapy sessions. This has been added to the discussion.

Page 11: 49% of potential participants lacked capacity and were ineligible. Bearing in mind that a very large proportion of people with #NOF will have cognitive issues, this has implications for generalisability of findings down the line. It would be useful to know the cognitive status of participants recruited into the feasibility study so as to truly understand who the study is feasible with/for.

REPLY: This has been added.

Page 18 first para. As you only measured FES-I cross-sectionally rather than longitudinally can you say that as fear of falling increased as ADLs decreased? Surely it is just that those with higher FoF have lower ADL activity??? ie an inverse relationship between FoF and ADLs – no surprise there though!!

REPLY: Yes, we agree. This has been amended.

General points:

I found the results section confusing to read as it kept jumping between recruitment and delivery issues so didn't flow as well as it could. Just needs reordering.

REPLY: This has been re-ordered.

If the Therapy manager software didn't really give you a clear picture of usual care across all sites is this going to be feasible moving forwards into the full trial process evaluation to use it. Not sure it is used much in England and I've never come across it before. Being able to clearly describe usual care is important and work we are currently doing suggests that if we can't clearly describe usual care in a

trial or it is so variable, it can be difficult to distinguish it from the intervention and may result in a null effect.

REPLY: Yes, we agree and in the phase III RCT we plan to use a patient-held treatment log to be completed by the visiting physio/OT/TL/nurse who is providing the usual rehabilitation care, which has been added to the implications section of the discussion.

You may also want to reference the findings of the recent CSP HipSPRINT national audit about usual care following hip fracture

REPLY: Thank you, this has been referenced and added to the discussion.

I like Box 1 on lessons learned but I think it could be expanded upon to include (a) the types of staff who would be trained to deliver the intervention and (b) alternative methods of collecting usual care data

REPLY: Thank you. Box 1 has been amended to include: Usual care recording; Patient-held treatment log completed by visiting therapists. We do not have enough evidence to say which staff should and should not be trained. As a pragmatic trial we believe that training should be kept flexible, depending on what is available and what works best at the different sites.

Please add CONSORT form for pilot trials <http://www.consort-statement.org/extensions/overview/pilotandfeasibility>

REPLY: This has already been done when reporting the main results of the feasibility study.

Reference: Williams NH, Roberts JL, Din NU, Totton N, Charles JM, Hawkes CA, et al. Fracture in the Elderly Multidisciplinary Rehabilitation (FEMuR): a phase II randomised feasibility study of a multidisciplinary rehabilitation package following hip fracture. *BMJ Open* 2016; 6: e012422.

FORMATTING AMENDMENTS (if any)

Required amendments will be listed here; please include these changes in your revised version:

1. Appendix File Format

Please re-upload your "Appendix file" under file designation supplementary files in PDF format.

REPLY: This has been done.

VERSION 2 – REVIEW

REVIEWER	Vicki Goodwin University of Exeter, UK
REVIEW RETURNED	22-May-2018

GENERAL COMMENTS	Thank you for addressing the reviewers comments. My only bug bear is in relation those with cognitive impairment and it must be explicit in the objectives and interpretation (abstract and main manuscript) that this intervention focussed on and was found to be feasible with cognitively intact people with hip fracture. Although interesting to note that one of the questions in the topic guide was about working with those with cognitive impairment - you may want to look at the qualitative work by Abi Hall (2017) in BMC Geriatrics on experiences of physios working with people with dementia and hip fracture.
---

REVIEWER	Lene Lindberg Karolinska Institutet, Sweden
REVIEW RETURNED	31-May-2018

GENERAL COMMENTS	I consider that the authors have responded thoroughly to the comments from the reviewers and made suggested changes.
--

VERSION 2 – AUTHOR RESPONSE

Reviewer: 3

It must be explicit in the objectives and interpretation (abstract and main abstract) that this intervention focussed on and was found to be feasible with cognitively intact people with hip fracture.

REPLY: This has been made more explicit in the abstract, strengths and limitations bullet points, aims and objectives, discussion.